



# Improvements to the Dynamic Wake Meandering Model by incorporating the turbulent Schmidt number

Peter Brugger[1], Corey Markfort[2], and Fernando Porté-Agel[1]

[1]Wind Engineering and Renewable Energy Laboratory (WiRE), École Polytechnique Fedérale de Lausanne (EPFL), 1015 Lausanne, Switzerland

[2]IIHR-Hydroscience & Engineering, Dept. of Civil and Environmental Engineering, The University of Iowa, Iowa City, IA 52242, USA

**Correspondence:** Peter Brugger (peter.brugger@epfl.ch)

**Abstract.** Predictions of the dynamic wake meandering model (DWMM) were compared to flow measurements of a scanning Doppler lidar mounted on the nacelle of a utility-scale wind turbine. We observed that the wake meandering strength of the DWMM agrees better with the observation, if the incoming mean wind speed is used as advection velocity for the downstream transport, while a better temporal agreement is achieved with an advection velocity slower than the incoming mean wind speed.

A subsequent investigation of the lateral wake transport revealed differences to the passive tracer assumption of the DWMM in addition to a non-passive downstream transport reported in earlier studies. We propose to include the turbulent Schmidt number in the DWMM to improve (i) the consistency of the model physics and (ii) the prediction quality. Compared to a benchmark, the thus modified DWMM showed a root-mean-square error reduction by 5% for mean velocity deficit and 7% for the turbulence intensity, relative to the unmodified DWMM, in addition to better temporal agreement of the dynamics. This is in contrast to an

error increase of 64% and 41% if only a more accurate downstream transport velocity is used without including the turbulent Schmidt number.

## 1   Introduction

Wind turbine wakes impinging on other wind turbines within a wind farm are a significant source of power losses and they decrease the lifetime of affected wind turbines. Wake meandering is a low-frequency horizontal and vertical oscillation of the

entire wake (Taylor et al., 1985). It affects power production due to its impact on the velocity deficit recovery, and it affects loads due to the turbulence added to the downstream flow (Larsen et al., 2013). Therefore, the modelling of wake meandering is one important aspect of wind farm development.

Modelling approaches for wake meandering can be grouped into two categories. The first group are computationally expensive large-eddy simulations (LES) that solve filtered flow equations at a high temporal and spatial resolutions (Mehta et al.,

2014). However, their high fidelity comes at the cost of a time consuming forward integration and a difficult initialisation of the simulation. The second group are computationally inexpensive engineering models like the dynamic wake meandering model (DWMM) (Larsen et al., 2008) and a statistical modelling approach (Thøgersen et al., 2017). The DWMM is based on the assumption that the wake behaves like a passive tracer, which is transported in the vertical and horizontal directions due to



**Incoming Flow**                 **Wind Turbine Wake**

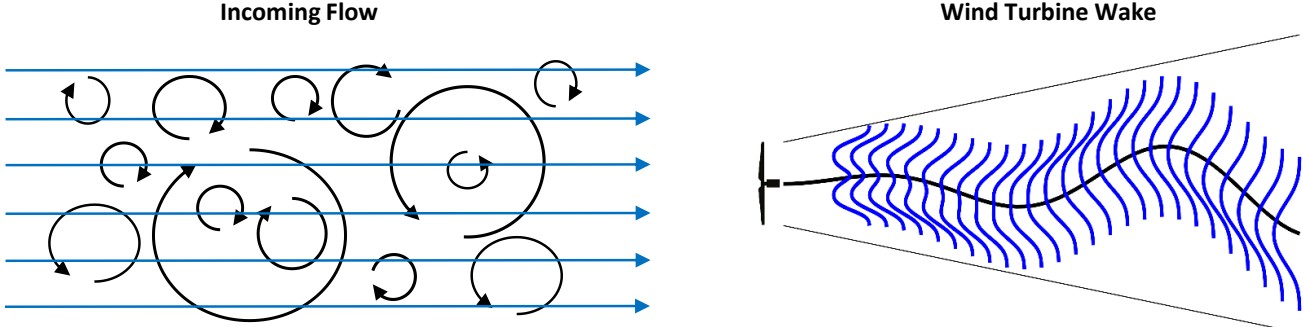

**Figure 1.** Illustration of wake meandering at an isolated wind turbine as assumed by the dynamic wake meandering model. Large-scale turbulence of the inflow displaces the wake of a wind turbine in the spanwise direction while it is transported downstream.

large-scale turbulence without (Fig. 1). The DWMM has the advantage of fast computation time and it can be initialized with
measurement data that are commonly available at a wind farm.

The DWMM model has seen validation efforts in literature, which are reviewed in the following. The underlying passive scalar assumption of the DWMM has been accepted with the exception of the downstream transport velocity of wake meandering, which is slower than the mean wind speed (Bingöl et al., 2010; Keck et al., 2014b; Machefaux et al., 2015; Conti et al., 2021; Brugger et al., 2022). Machefaux et al. (2015) additionally investigated the lateral transport velocity of the wake
while it is meandering, but they had no measurements of the lateral velocity of the inflow for comparison. A validation of the mean velocity field and turbulence intensity predicted by the DWMM against field measurements showed good agreement and revealed a sensitivity to the eddy-viscosity parametrisation used (Reinwardt et al., 2018, 2020).

The above-mentioned discrepancies between the passive tracer assumption of the DWMM and observed transport behaviour warrant closer examination. Specifically, assuming the wake as a passive tracer in the cross-stream directions and non-passive
in the streamwise direction is physically inconsistent. Also, an investigation of the impact of the downstream advection velocity on the predictions of the DWMM has not been carried out so far. Further, previous validation efforts for the velocity deficit and the turbulence intensity predicted by the DWMM focused on validating all components of the DWMM simultaneously, with the exception of Reinwardt et al. (2020). Here, we extend the validation of the wake meandering module of the DWMM by Reinwardt et al. (2020) to the turbulence intensity. This is especially interesting, because the meandering framework of the
DWMM is what sets it fundamentally apart from analytical wake models.

Therefore, this paper will compare the wake dynamics modelled by the DWMM to the wake dynamics observed with field measurements and, further, how it affects the predictions of the DWMM for the effect of wake meandering on the mean velocity deficit and the turbulence intensity. These research questions will be studied across a wide range of atmospheric conditions using field measurements of two pulsed Doppler LiDARs at a utility-scale wind turbine.



## 2 Methods

This section introduces first the DWMM (Sect. 2.1) followed by the research site and the instrument setup (Sect. 2.2) with which the data set for the model validation was collected (Sect. 2.3).

### 2.1 Dynamic wake meandering model

The dynamic wake meandering model (DWMM) was introduced by Larsen et al. (2007, 2008) and assumes the wake as a passive tracer that is advected by the large-scale turbulence of the atmospheric boundary layer. The model decomposes the wake into three parts: (i) a quasi-steady velocity deficit calculated with the thin shear layer approximation of the Naviar-Stokes equations (Ainslie, 1988), (ii) a wake meandering part modelled as a displacement of the entire wake with the large-scale turbulence of the background flow (Larsen et al., 2008), and (iii) small-scale turbulence that is superimposed on the flow field. A schematic illustation of part (i) and (ii) is on the right of Fig. 1 with the quasi-steady velocity deficit in blue and the wake displacement in black. We follow here the implementation of Reinwardt et al. (2020) for the quasi-steady velocity deficit, including their recalibration, and Bingöl et al. (2010) for the wake meandering part. The small-scale turbulence part of the DWMM is not required here, because the present investigation focuses on the wake meandering part of the DWMM. The DWMM was implemented using the commercial software Matlab for this study.

#### 2.1.1 Quasi-steady velocity deficit

The quasi-steady velocity deficit is modelled with the steady-state, axisymmetric thin shear layer approximation of the Naviar-Stokes equations with an eddy-viscosity turbulence closure (Ainslie, 1988). The momentum equation is given by

$$\overline{u}\frac{\partial\overline{u}}{\partial x} + \overline{v}_r\frac{\partial\overline{u}}{\partial r} = \frac{1}{r}\frac{\partial}{\partial r}\left(\nu r\frac{\partial\overline{u}}{\partial r}\right) \tag{1}$$

and the continuity equation is given by

$$\frac{1}{r}\frac{\partial}{\partial r}(r\overline{v}_r) + \frac{\partial\overline{u}}{\partial x} = 0, \tag{2}$$

where $\overline{u}$ is the mean wind speed in the axial direction, $\overline{v}_r$ is the mean wind speed in the radial (or spanwise) direction, $r$ is the radial coordinate, $x$ is the downstream coordinate, and $\nu$ is the eddy viscosity. The recalibrated mixing-length parametrisation of the eddy viscosity of Reinwardt et al. (2020) is given by

$$\frac{\nu}{\overline{u}_{hub}R} = k_1 F_1(\tilde{x})TI_u + k_2 F_2(\tilde{x})\max\left(\frac{R_w(\tilde{x})^2}{R\overline{u}_{hub}}\left|\frac{\partial U(\tilde{x})}{\partial r}\right|, \frac{R_w(\tilde{x})}{R}\left(1 - \frac{\overline{u}_{min}(\tilde{x})}{\overline{u}_{hub}}\right)\right), \tag{3}$$

where $\overline{u}_{hub}$ is the mean wind speed at hub height, $\overline{u}_{min}$ is the minimum velocity of the wake, $TI_u$ is the longitudinal turbulence intensity at hub height, $\tilde{x} = x/R$, $R$ is the rotor radius, $R_w$ is the wake width, $k_1 = 0.0914$ and $k_2 = 0.0216$ are calibration constants, and $F_1$ and $F_2$ are empirical filter functions given by

$$F_1 = \begin{cases} 0.25\tilde{x}, & \text{for } \tilde{x} < 4, \\ 1, & \text{for } \tilde{x} \geq 4, \end{cases} \tag{4}$$





and

$$F_2 = \begin{cases} 0.035, & \text{for } \tilde{x} < 4, \\ 1 - 0.965 \exp\left(-0.35(0.5\tilde{x} - 2)\right), & \text{for } \tilde{x} \geq 4. \end{cases} \tag{5}$$

The system of partial differential equations given by Eq. (1) and Eq. (2) was solved on an isotropic grid with a resolution of $0.01D$ spanning $10D$ from the origin at the nacelle using the method of Crank and Nicolson (1947). The inner boundary condition is $V(r = 0) = 0$, and the outer boundary condition is $U(r = 10D) = u_{hub}$. The initial condition at $x = 0$ is introduced in the next section.

### 2.1.2   Initial velocity deficit at the rotor plane

The thin shear layer equations (Eq. 1 and Eq. 2) omit the pressure gradient terms. The effect of the pressure gradients is considered negligible at a distance of $3D$ (Madsen et al., 2010). Therefore, the boundary condition at $x = 0$ is designed to account for the effect of the neglected pressure gradient by including expansion and deceleration of the flow at the rotor disc such that the resulting flow field after $3D$ is accurately represented. The initial velocity deficit is iteratively given by

$$\overline{u}_{ini}\left(r_{w,i}\right) = \overline{u}_{hub}(1 - (1 + f_u)\overline{a}) \tag{6}$$

and

$$r_{w,i} = r_i \sqrt{\frac{1 - \overline{a}}{1 - (1 + f_R)\overline{a}}} \tag{7}$$

with $f_u = 1.1$, $f_R = 0.98$, $\overline{a}$ is the induction factor averaged along all radial positions, $r_i$ is the rotor radius at position $i$, and $r_{w,i}$ is the wake radius at position $i$ (Keck et al., 2013). The induction factor is computed from the thrust coefficient of the wind turbine ($C_T$) by using the relationship

$$C_T = 4\overline{a}(1 - \overline{a}), \tag{8}$$

where $\overline{a}$ is assumed to be constant across the rotor area. The assumption is necessary because the radial distribution of the induction factor is not available to us for the wind turbine at the research site. However, testing with two different induction factor distributions of model wind turbines shown in literature (scaled so that they yield the same mean induction factor) showed that the two initial velocity deficits at $x > 4D$ had a mean absolute difference of 0.9% with a maximum of 1.5% based

on the mean wind speed for the wind speed range covered in the results.

### 2.1.3   Wake meandering

The DWMM uses the hypothesis that the wake can be modelled as a passive tracer that is transported by the large-scale turbulence structures of the atmospheric boundary layer. The process can be imagined as a continuous sequence of velocity deficits emitted by the wind turbine that are passively transported by the large-scale turbulence (Larsen et al., 2008). Therefore,



a suitable description of the turbulence field is required. We depart here from the implementation of Reinwardt et al. (2020), who used a Kaimal spectrum to generate a stochastic turbulence field, and instead we will follow the approach of Bingöl et al. (2010) that is more suitable for a direct comparison with wake measurements. They adopted Taylor's frozen turbulence hypothesis (Taylor, 1938) and assumed that the large-scale turbulence is correlated across the rotor area. The instantaneous wake position is then given by

$$\frac{\mathrm{d}x_{pre}(t, \Delta T)}{\mathrm{d}t} = \overline{u}_a, \tag{9}$$

$$\frac{\mathrm{d}y_{pre}(t, \Delta T)}{\mathrm{d}t} = v(t, \Delta T), \tag{10}$$

and

$$\frac{\mathrm{d}z_{pre}(t, \Delta T)}{\mathrm{d}t} = w(t, \Delta T), \tag{11}$$

where the subscript "pre" stands for prediction, $\overline{u}_a$ is the downstream advection velocity (also called downstream transport velocity), $v$ and $w$ are the large-scale lateral and vertical turbulent velocity fluctuations, $t$ is the time of velocity deficit "emission", and $\Delta T$ is the time that has elapsed since a specific velocity deficit "emission". Using the lateral velocity at the turbine location for the right-hand side of Eq. (10), the instantaneous wake center position in the horizontal plane is given by

$$y_{pre}(x, t) = v(t - \Delta T(x))\Delta T(x), \tag{12}$$

where $\Delta T$ was expressed as a function of the downstream distance $x$ with

$$\Delta T(x) = \int_0^x \frac{dx}{\overline{u}_a(x)}. \tag{13}$$

We will compare two assumptions for the downstream advection velocity in the results: (1) the advection velocity is the same as mean wind speed with $\overline{u}_a(x) = \overline{u}_{hub}$, and (2) the advection velocity is given by the average of the mean wind speed and mean velocity at the wake center with

$$\overline{u}_a(x) = 0.5(\overline{u}_{cen}(x) + \overline{u}_{hub}) \tag{14}$$

as proposed by Cheng and Porté-Agel (2018). For the latter, the mean velocity at the wake center $\overline{u}_{cen}(x)$ is computed with the analytical wake model of Bastankhah and Porté-Agel (2016) (see Eq. (A2) in appendix B). Assumption (1) is following the simplified DWMM in Larsen et al. (2008). Assumption (2) is an improvement on Keck (2015), who assumed that the downstream advection velocity is a constant 80% of the mean wind speed.

The vertical component of wake meandering cannot be computed directly, because measurements for the right-hand side of Eq. (11) are not available. Instead, we assume that the vertical wake meandering can be modelled to be proportional to the lateral wake meandering with

$$z_{pre}(x, t) = r_{yz} y_{pre}(x, t), \tag{15}$$





where the factor $r_{yz}$ is the ratio between the horizontal and the vertical wake meandering strength.

We assume that a suitable choice of $r_{yz}$ is the ratio of lateral to vertical turbulence intensity. For a purely shear driven ABL, ratios of $TI_v/TI_u$ and $TI_w/TI_v$ are about $0.5$, while for a purely convective ABL $TI_v \approx TI_u < TI_w$ above the surface layer and $TI_v \approx TI_u > TI_w$ within the surface layer (Moeng and Sullivan, 1994). Wind tunnel experiments with purely shear-driven flows showed that wake meandering in the vertical direction had a smaller amplitude than the lateral direction (España et al., 2012; Bastankhah and Porté-Agel, 2017), which supports the assumed proportionality to the turbulence intensity ratios. Keck

et al. (2014a) presented ratios for vertical to lateral wake meandering of approximately $0.6$, $0.8$, and $0.9$ for stable, neutral, and unstable conditions, respectively. We do not have direct measurements of $TI_w/TI_v$ and the data available to us from a nearby meteorological tower was not suitable to determine the boundary layer state. Therefore, we assume $r_{yz} = 0.8$ as an average ratio.

Further, Eq. (15) implies a perfect correlation between $y_{pre}$ and $z_{pre}$, which might not be the case in reality. However,

randomly rearranging $z_{pre}$ changed the slopes of the linear regressions shown in Sect. 3.2 by less than $0.02$ (no detectable change for the intercept and the correlation coefficient) and, therefore, does not affect the drawn conclusions.

## 2.2 Research site and measurement setup

The research site and the measurement setup is the same as reported in Brugger et al. (2022). The setup was implemented between 19 August 2017 and 2 October 2017. Quality assurance and data selection criteria are summarized at the beginning of

Sect. 3.

The site consists of an isolated wind turbine located at Kirkwood Community Collage in Cedar Rapids, Iowa (Fig. 2). The wind turbine is a 2.5 MW Liberty C96 from Clipper Windpower with a hub height of $z_{hub} = 79$ m and a rotor diameter of $D = 96$ m. The area in the vicinity of the wind turbine is urbanized with some agricultural farmland to the south and east. Data from the supervisory control and data acquisition (SCADA) system of the wind turbine are available to us with a 10-minute

resolution.

We installed two pulsed Doppler LiDARs on the roof of the nacelle (Fig. 3a). A Doppler LiDAR emits infrared laser pulses that are scattered by aerosols within the atmosphere. The backscattered light receives a frequency shift due to the Doppler effect caused by the movement of the aerosols. Assuming that aerosols are transported by the wind, the line-of-sight velocity of the air along the laser beam can be estimated from the frequency shift detected by the instrument. The instruments were Stream

Line models from Halo Photonics Ltd. (Worecestershire, UK). The instruments were programmed to sample the velocity with a temporal frequency of 3 Hz and spatial resolution of 18 m along the laser beam.

The Doppler LiDAR mounted towards the rear of the nacelle was programmed to perform 230 successive Plan Position Indicator (PPI) scans in the downstream direction with an opening angle of $\pm 12°$ and an azimuth step of $2°$ to capture the wake (Fig. 3). Each individual PPI sweep with the return to the starting position took 7.2 s and the full scan was completed

in approximately 28.4 minutes. The scans were designed to be slightly shorter than the 30 minute period allocated in the scan schedule to ensure smooth operation of the measurements. Simultaneously, the front mounted Doppler LiDAR was programmed to measure the lateral velocity component with a horizontal, fixed beam at a $90°$ to the rotor axis for a period of





**Figure 2.** Satellite image of the measurement site with the location of the wind turbine (© Google Earth). The wind turbine coordinates are $41.9165°$ latitude and $-91.6508°$ longitude.

14 minutes (for the remainder of the 30 minute period, it measured with a fixed beam parallel to the rotor axis upstream, but those measurements are not used here). This scan pattern of the two Doppler LiDARs was scheduled to begin every second
165  hour at the half hour mark.

### 2.3  Model inputs and reference data

The input values of the DWMM and the reference of the validation use data from separate measurement instruments to keep them independent. The model input is generated from the SCADA data and the measurements of the front-mounted Doppler LiDAR. The measurements of the rear-mounted Doppler LiDAR are used as the reference for the validation.





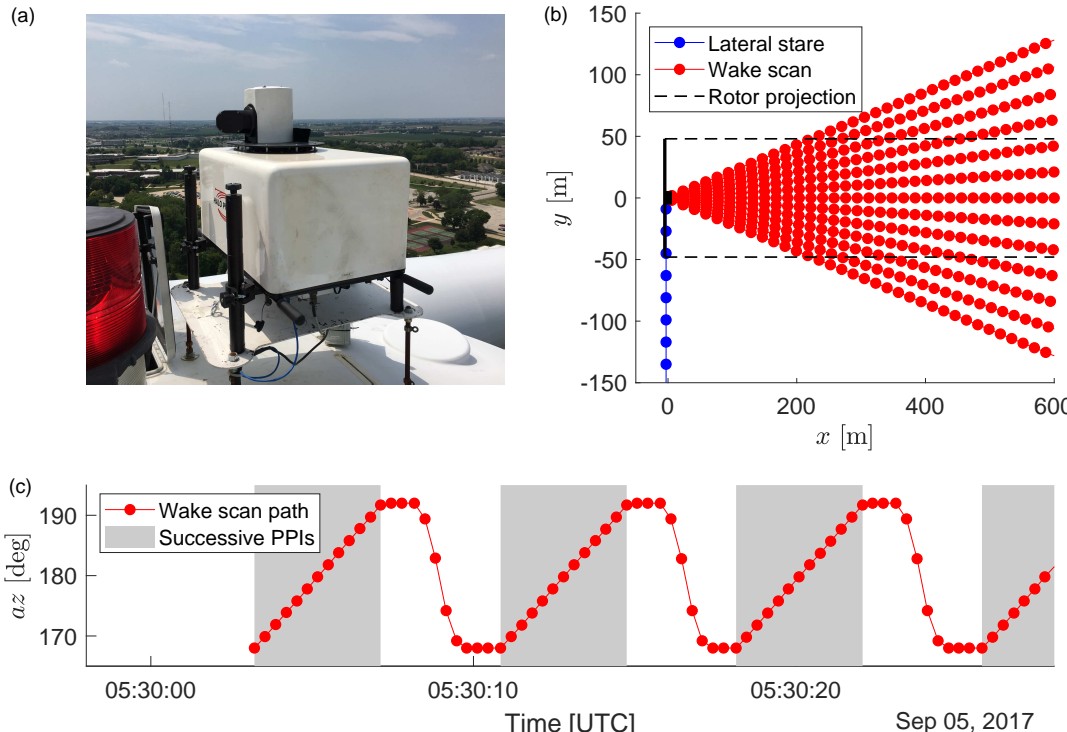

**Figure 3.** Photo of the front-mounted Doppler LiDAR on the nacelle of the wind turbine (a). Scan patterns of the nacelle mounted Doppler LiDARs viewed from top (b). Wake scans of the rear-mounted Doppler LiDAR (red) were accompanied by measurements in a lateral staring mode of the front-mounted Doppler LiDAR (blue). LiDAR beams are shown as lines with range gate centers indicated as points. The wind turbine is stylized in black and the rotor-edge projection in the wind direction is indicated with black dashed lines. The bottom panel shows the scanner path for a section of a wake scan (c), where the grey area indicates the successive PPIs that, to-gether, become a wake scan. Figure adapted from Brugger et al. (2022) with changes under the Creative Commons Attribution 4.0 License (https://creativecommons.org/licenses/by/4.0/).

### 2.3.1 Model inputs

The input variables of the DWMM and the measurements from which they are taken are listed below:

- $\overline{u}_{hub}$ is the mean wind speed at hub height. It is measured by a cup anemometer located on the roof of the wind turbine and reported in the SCADA data. Because the SCADA data has a 10-minute resolution, the averaging period of the mean wind speed is longer than the 14-minute measurement period of the front-mounted Doppler LiDAR.

- $v(t)$ is the time series of the large-scale lateral velocity of the inflow. It is generated from the measurements of the front-mounted Doppler LiDAR using the range gate at a distance of 117 m, which is the closest range gate that is not affected by the rotor. A linear trend is removed and a low-pass filter with a threshold of $\beta \Delta T$ with $\beta = 0.8$ is applied to isolate the large-scale turbulence fluctuations (Cheng and Porté-Agel, 2018). This threshold of the low-pass filter is



considerably larger than the rotor diameter and it is therefore reasonable to assumed that $v(t)$ is representative for the full rotor area. A trend is removed from $v(t)$ instead of the mean to remove non-stationary effects, like a steady change of the wind direction over a 14-minute period, that are not considered wake meandering. Further, the lateral turbulence intensity $(TI_v)$ and the integral time scale $(T_{i,v})$ are computed from $v(t)$ prior to the low-pass filtering. A proportionality between longitudinal and lateral turbulence intensity with $TI_u = \frac{3}{2}TI_v$ is assumed.

– $C_T$ is the wind speed dependent thrust coefficient of the wind turbine. It is selected from the thrust curve in Fig. A1 based on $\overline{u}_{hub}$.

### 2.3.2 Reference data set

The instantaneous wake center position, the turbulence intensity added by wake meandering, and the reduction of the mean velocity deficit due to wake meandering are extracted from the measurements of the rear mounted Doppler LiDAR. The processing steps listed below and illustrated in Fig. 4 are similar to those in Brugger et al. (2022):

1. A signal-to-noise ratio (SNR) threshold of $-17$ dB is used to reject low quality Doppler LiDAR measurements (Pearson et al., 2009).

2. The remaining line-of-sight velocities are gridded on a polar coordinate system $u_r(\phi, r, t)$ with an azimuth $(\phi)$ resolution of $2°$, a radial $(r)$ resolution of $18$ m, and a time step $(t)$ aligning with the PPIs of the wake scans. The $az$ positions of the LiDAR scans and the $\phi$ positions of the polar coordinate system can have a difference of $0.2°$ towards the end of a PPI resulting from the acceleration phase of the scanner head, small variations of the scanner behaviour, and fluctuations of the sampling frequency. Multiple measurements are available for the outside grid points due to a short resting time of the scanner at the turn-around point and the measurements closest in time are used at those grid points.

3. The transformation to a Cartesian coordinate system is made using $y = r\sin^{-1}(\phi)$ and $x = \langle r\cos^{-1}(\phi)\rangle$, where the angle brackets indicate the lateral averaging.

4. An instantaneous velocity deficit is computed with $\Delta u_r(x, y, t) = \overline{u}_{hub} - u_r(x, y, t)$. This approach maps all measurements during a PPI on a single time stamp neglecting the travel time of the scanner head.

5. The instantaneous position of the wake center is detected with the centroid of the velocity deficit with

$$y_{wc}(x,t) = \frac{\sum_y y\Delta u_r(x,y,t)}{\sum_y \Delta u_r(x,y,t)}, \tag{16}$$

where the subscript "wc" stands for wake centroid.

6. The instantaneous velocity deficit in the meandering frame of reference $(\Delta u_r(x, \tilde{y}, t))$ is computed by transformation of the lateral coordinate with $\tilde{y} = y - y_{wc}$ and interpolated on the original $y$ grid points.

**Figure 4.** The data processing steps to quantify the effect of wake meandering on the mean velocity deficit and the turbulence intensity are illustrated for an example at a downstream distance of $x = 5D$. The instantaneous velocity deficit in the nacelle frame of reference (a) and meandering frame of reference (b) have a temporal average (c, e) and standard deviation (d, f) applied. The reduction of the mean velocity deficit due to wake meandering is then quantified as amplitude difference (g) of two Gaussian fits to the two velocity deficits in (c) and (e). The turbulence intensity added by wake meandering is quantified as the spatially averaged difference in turbulence intensity (h).





7. The temporal mean and the standard deviation of $\Delta u_r(x,y,t)$ and $\Delta u_r(x,\tilde{y},t)$ provide profiles of the mean velocity deficit and the turbulence intensity in the nacelle frame of reference (NFOR) and the meandering frame of reference (MFOR), respectively. The NFOR is identical to a fixed frame of reference here, because data was selected to have no yaw activity of the wind turbine.

8. The reduction of the mean velocity deficit due to wake meandering is then quantified by the amplitude difference of two Gaussian functions fitted to the mean velocity deficit in the NFOR and the MFOR, respectively. The difference will be denoted as $\widetilde{C} - C$ where $C$ ($\widetilde{C}$) is the amplitude of the Gaussian function in the NFOR (MFOR).

9. The turbulence added by wake meandering is quantified by the laterally averaged difference in turbulence intensity between the NROR and the MFOR. It will be denoted as $\langle TI_u - \widetilde{TI}_u \rangle$.

10. Lastly, a temporal linear trend is also removed $y_{wc}(x,t)$ and a low-pass filter is applied to make it comparable to $y_{pre}(x,t)$ of the DWMM that is based on the detrended and low-pass filtered $v(t)$. Note that $\widetilde{C} - C$ and $\langle TI_u - \widetilde{TI}_u \rangle$ still include non-stationary effects and we will deal with that in Sect. 3.2.1.

The above processing steps are applied for downstream distances between $xD^{-1} = 3$ and $xD^{-1} = 7$. The Doppler lidar's field of view and the double-peak shape of velocity deficit in the near-wake were problems for the detection of $y_{wc}$ for $xD^{-1} < 3$, and the lateral resolution of the lidar scans became coarser than 10 m for $xD^{-1} > 7$. Some parts of the results will focus on a downstream distance of $x = 5D$, because the scanning cone has an ideal width of $2D$ there and $5D$ is typical distance between wind turbines in onshore wind farms.

## 3 Results

The first part of the results will focus on the modelling of wake meandering itself, while the second part will focus on the validation of the predicted effects that wake meandering has on the mean velocity deficit and the turbulence intensity. A set of 43 cases, each covering an approximately 14-minute period, was selected from the measurement data of the campaign. The selection criteria were a mean wind speed above $5 \text{ m s}^{-1}$, no yaw movement of the wind turbine, a sufficient mean SNR of the Doppler lidar, and a wake within the Doppler lidar's scanning cone. The 43 cases cover a wind speed range from $5 \text{ m s}^{-1}$ to $11 \text{ m s}^{-1}$, turbulence intensities of the lateral velocity component up to $8\%$ (for higher turbulence intensities, the wake is not covered by the Doppler LiDARs scanning cone due to very strong wake meandering). The selection criteria and the data set are identical to Brugger et al. (2022).

### 3.1 Testing of the passive scalar assumption of the DWMM

First, the predictions of the instantaneous wake center positions will be validated. The root-mean-square error (RMSE) and the correlation coefficient between the predicted wake center position (Eq. 12) and the observed wake center position (Eq. 16) will be used as quality metrics. The observed wake center position has been detrended and low-pass filtered with the same filter





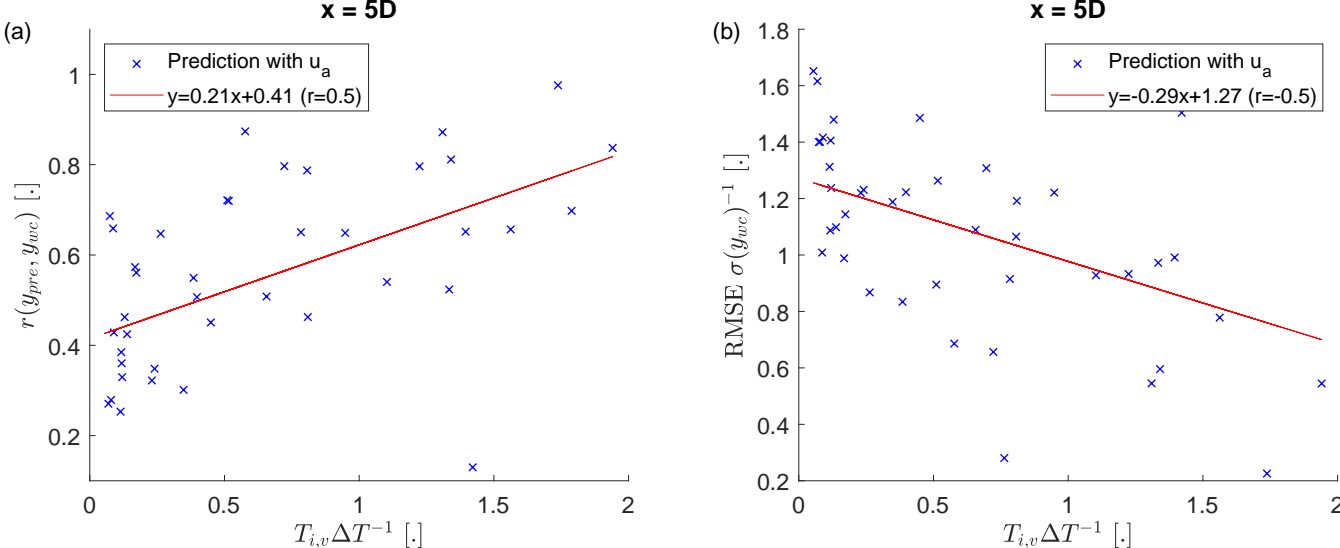

**Figure 5.** The correlation coefficient (a) and normalized root-mean-square error (b) between the wake center position predicted by the dynamic wake meandering model ($y_{pre}$, Eq. 12) and the observed wake position by the wake scanning Doppler lidar ($y_{wc}$, Eq. 16) at a downstream distance of $5D$ from the wind turbine. The ratio between the integral time scale of lateral velocity ($T_{i,v}$) and the time delay due to downstream advection ($\Delta T^{-1}$) quantifies the rate of evolution of the turbulent wind field during the time of downstream advection.

threshold as the input of the DWMM for comparability. The results of the evaluation at $x = 5D$ are shown in Fig. 5. They have a general trend of an increasing correlation and a decreasing normalized RMSE with $T_{i,v}\Delta T^{-1}$. The ratio $T_{i,v}\Delta T^{-1}$ quantifies the rate of evolution of the turbulent wind field during the time of downstream advection (or in other words how

well the Taylor's frozen turbulence hypothesis holds). This behaviour is in agreement with the recommendation that for large downstream distances, the spatial variability of the large-scale turbulence components should be taken into account (Larsen et al., 2008), which is supported by the results of a previous data analysis (Brugger et al., 2022).

Next, we will investigate the effect of the downstream advection velocity on the predicted wake-center positions. We will compare predictions using the reduced downstream advection velocity given by Eq. (14) with predictions using the mean wind

speed of the inflow as the downstream advection velocity. The effect of the choice of advection velocities on the correlation coefficient is shown in Fig. 6a. Using the reduced advection velocity $\overline{u}_a$ improved the correlation between predictions and observations for $x < 5D$, indicating better temporal alignment for the predictions using the reduced downstream advection velocity. No systematic effect on the correlation is observed beyond $5D$, which can be explained by $\overline{u}_a(x)$ approaching $\overline{u}_{hub}$ with increasing $x$ and the decorrelation of the two turbulent signals with increasing separation. Despite the increase in correlation

coefficient, using $\overline{u}_a$ had a detrimental effect on the normalized RMSE (Fig. 6b). This will be investigated in more detail in the following section.





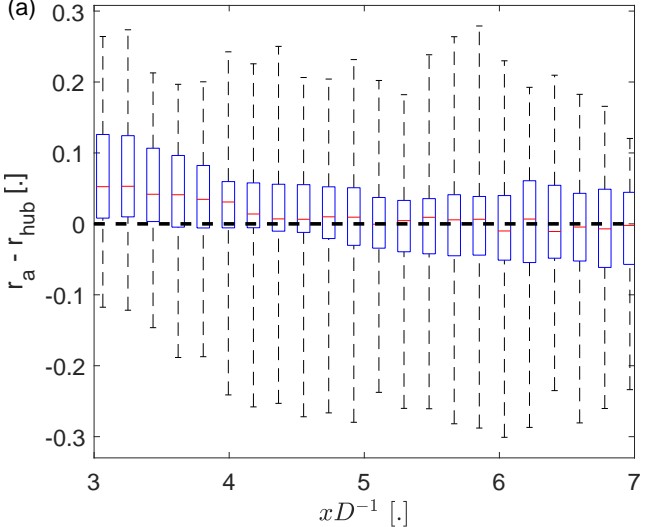
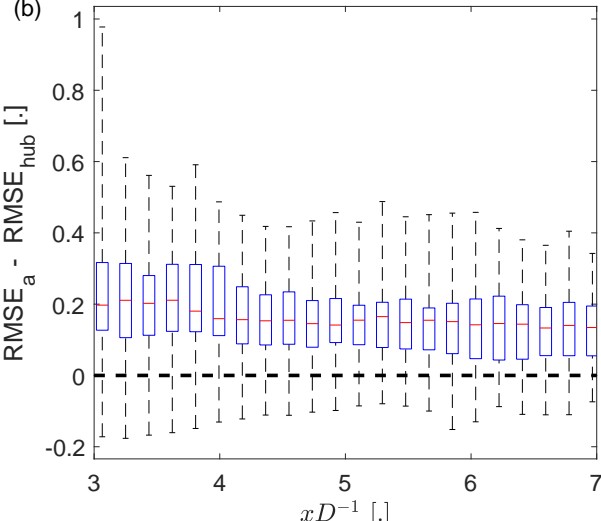

**Figure 6.** The effect of the downstream advection velocity on the correlation coefficient (a) and on the normalized root-mean-square error (b) between $y_{pre}$ and $y_{wc}$ as a function of the downstream distance. The subscript "a" ("hub") indicates $\overline{u}_a$ ($\overline{u}_{hub}$) as downstream advection velocity. The whiskers show the range of the data, the top and bottom of the blue box indicating the 25th and 75th percentile, and the red center marker showing the median.

### 3.1.1 Overestimation of the wake meandering strength

It was previously observed that using $\overline{u}_a$ as the advection velocity increases the RSME despite having a higher correlation with the observation. Analysing several cases visually, we observed that the predictions had in many cases a too large amplitude of the wake displacement compared to the observations. Figure 7 shows three examples from the data set to illustrate the behaviour. It is apparent that the better temporal alignment of the predictions using $\overline{u}_a$ is accompanied by an overestimation of the wake displacement compared to the predictions with $\overline{u}_{hub}$. Quantifying the overestimation with the difference in wake meandering strength for the whole data set, it becomes clear that this is a systematic bias that is introduced by the reduced advection velocity (compare Fig. 8a and Fig. 8b).

Better temporal agreement of the DWMM when using a downstream advection velocity slower than the mean wind speed was observed in several studies (Bingöl et al., 2010; Keck et al., 2014b; Machefaux et al., 2015). However, a subsequent overestimation of the wake meandering strength has not been reported as a problem so far. While Keck et al. (2014a) showed in their Fig. 3 that the lateral wake meandering strength predicted by the DWMM increases with a slower downstream transport velocity for $x < 10D$, they did not further investigate the matter (possibly due to a scaling of the wake meandering strength at a later stage of their DWMM implementation). In a previous validation of the passive scalar assumption by Bingöl et al. (2010), this phenomenon was not reported, but a visual inspection of their Fig. 8 suggests that three of their four cases also exhibit a larger displacement of the wake center position in the DWMM predictions compared to the wake measurements. Trujillo et al. (2011) showed an example case in their Fig. 5 which we digitized (linearly interpolating the parts that where obstructed




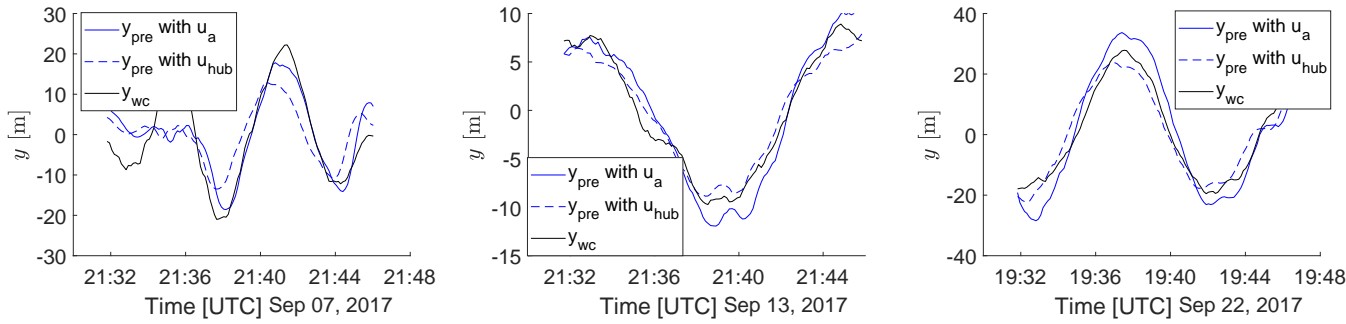

**Figure 7.** The time series of the observed and the predicted wake center positions at $x = 5D$ for three example cases, which were selected for their high correlation between reference and prediction. The predictions are shown for $\overline{u}_{hub}$ (dashed blue) and $\overline{u}_a$ (solid blue) as the downstream transport velocity. The reference from the observations is shown in black.

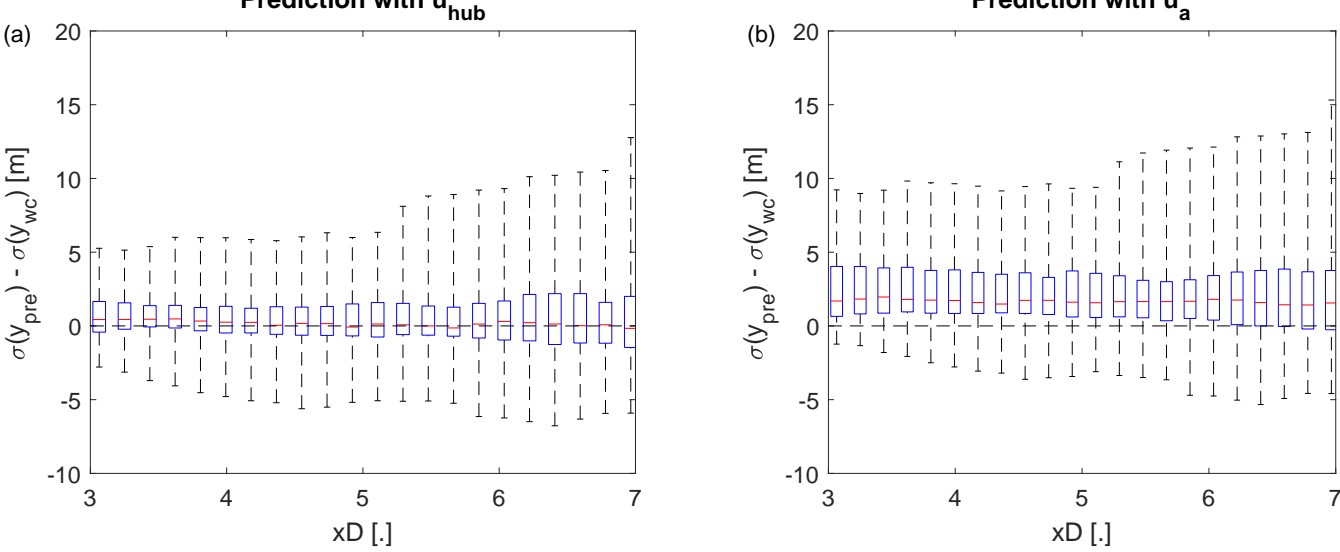

**Figure 8.** Error between observed and predicted wake meandering strength with $\overline{u}_{hub}$ (a) and $\overline{u}_a$ (b) as downstream advection velocity. The whiskers show the range of the data, the top and bottom of the blue box indicate the 25th and 75th percentile, and the red center marker shows the median.





by other plot elements) and found similar displacements of the wake center position for the predictions and the observations.

However, the description of their data processing mentions low-pass filtering only for the modelled wake of the DWMM and not the observed wake, which could mask the overestimation. Other validations might not have observed this issue previously, because using the mean wind speed for the downstream advection and a temporally averaged validation approach masks the issue (Reinwardt et al., 2018, 2020; Conti et al., 2021).

Based on our own findings and the literature review, we believe that the discrepancy between temporal agreement and wake

meandering strength points towards a short-coming of the passive scalar assumption of the DWMM. In the following section, we will provide a hypothesis to addresses this problem. Other possible explanations for the overestimation that were tested on the data and rejected are listed in Appendix C.

### 3.1.2 Improvement of the DWMM to account for momentum transport

We hypothesize that the transport of the wake with large-scale turbulence is more akin to the transport of momentum than the

transport of a passive tracer and that, subsequently, the turbulent Schmidt number should be considered in the modelling of wake meandering. The turbulent Schmidt number characterises the ratio between the turbulent transport of momentum and the turbulent transport of passive scalars. Previous experiments indicated that momentum is transported less efficiently than scalars in turbulent wakes (Reynolds, 1976; Antonia et al., 1993). First, we provide support for this hypothesis by comparing observed transport behaviour of the wake with the expectation of a passive scalar. Then we include the turbulent Schmidt number into

the DWMM and compare those new predictions with the observations.

We use the diffusion theory of Taylor (1922) to compare the observed transport of wake meandering with the expected transport of a scalar. Cheng and Porté-Agel (2018) adapted the diffusion theory from a point source to an area source for wind turbine wakes. The standard deviation of a lateral profile of a scalar concentration in a wake at $\Delta x$ downstream of the virtual point source is then given by

$$\sigma_{y,scalar} = \left\langle v^2 \right\rangle^{0.5} \frac{\Delta x}{\overline{u}_a}, \tag{17}$$

where $\left\langle v^2 \right\rangle^{0.5}$ is the standard deviation of the lateral air velocity. If momentum is considered, which is transported less efficient than a scalar in a wake (Reynolds, 1976), this expression becomes

$$\sigma_{y,wake} = \sqrt{Sc_t} \left\langle v^2 \right\rangle^{0.5} \frac{\Delta x}{\overline{u}_a}, \tag{18}$$

where $Sc_t$ is the turbulent Schmidt number. We assume that the standard deviation of the lateral transport velocity of the

wake centroid can be expressed as $\left\langle v_{wake}^2 \right\rangle^{0.5} = \sqrt{Sc_t} \left\langle v^2 \right\rangle^{0.5}$. With this assumption, we can determine the turbulent Schmidt number of wake meandering from the ratio of the standard deviation of the lateral velocity to the standard deviation of the lateral velocity of the wake centroid:

$$Sc_t = \left( \left\langle v_{wake}^2 \right\rangle^{0.5} \left\langle v^2 \right\rangle^{-0.5} \right)^2. \tag{19}$$

The lateral transport velocity of the wake center ($v_{wake}$) can be determined from the lateral displacement of the wake center

position following a method of Machefaux et al. (2015). First, the time lag $\Delta t$ between $v(t)$ and the wake center position $y_{wc}(t)$



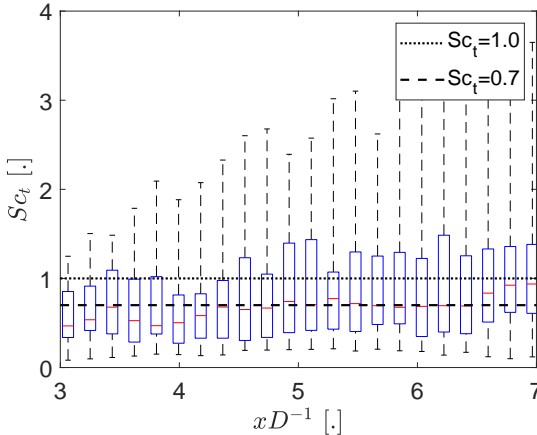

**Figure 9.** Turbulent Schmidt numbers of wake meandering (Eq. 19) as function of the downstream distance from the wind turbine. The whiskers show the range of the data, the top and bottom of the blue box indicate the 25th and 75th percentile, and the red center marker shows the median.

for a given downstream distance is determined with a cross-correlation. We do not use Eq. (13) to compute the time delay to make no assumptions on the downstream transport velocity here. Then, the lateral transport velocity of the wake center is estimated with $v_{wake} = y_{wc}/\Delta t$. If the cross-correlation for determining $\Delta t$ is lower than $0.8$ the estimate of $v_{wake}$ is rejected.

The turbulent Schmidt number was determined with Eq. (19) using $v$ measured by the front-mounted Doppler lidar as described in Sect. 2.3.1 and $v_{wake}$ determined as described in the previous paragraph. Both time series were low-pass filtered and detrended to make them comparable. The results in Fig. 9 show that the turbulent Schmidt numbers of wake meandering are smaller than unity for the majority of the data set. The average of the observed $Sc_t$ is $0.68$, which close to $0.7$ used by Cheng and Porté-Agel (2018) for wind turbine wakes based on Reynolds (1976). This suggests that wake meandering is more akin to the transport of momentum than the transport of a passive scalar. Using an average over multiple range gates to determine $v$ to

mirror the spatial averaging of the wake center detection changes the average $Sc_t$ to $0.69$. Down sampling $v$ to the temporal resolution of $y_{wc}$ prior to low-pass filtering has also only a small effect on the results by changing the average $Sc_t$ to $0.64$.

Following our hypothesis, we modified the DWMM to account for a reduced momentum transport efficiency by including $\sqrt{Sc_t}$ with constant $Sc_t = 0.7$ in Eq. (12). The results are shown in Fig. 10. Including $Sc_t$ reconciled $81\%$ of the overestimation of the wake meandering strength.



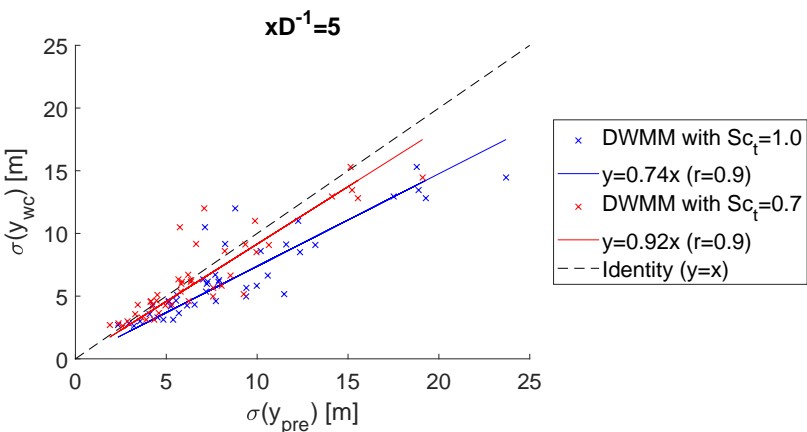

**Figure 10.** Observed wake meandering strength and predicted wake meandering strength of the DWMM. Blue indicates the DWMM with the passive scalar assumption and red indicates the DWMM modified with turbulent Schmidt number to account for a less efficient momentum transport.



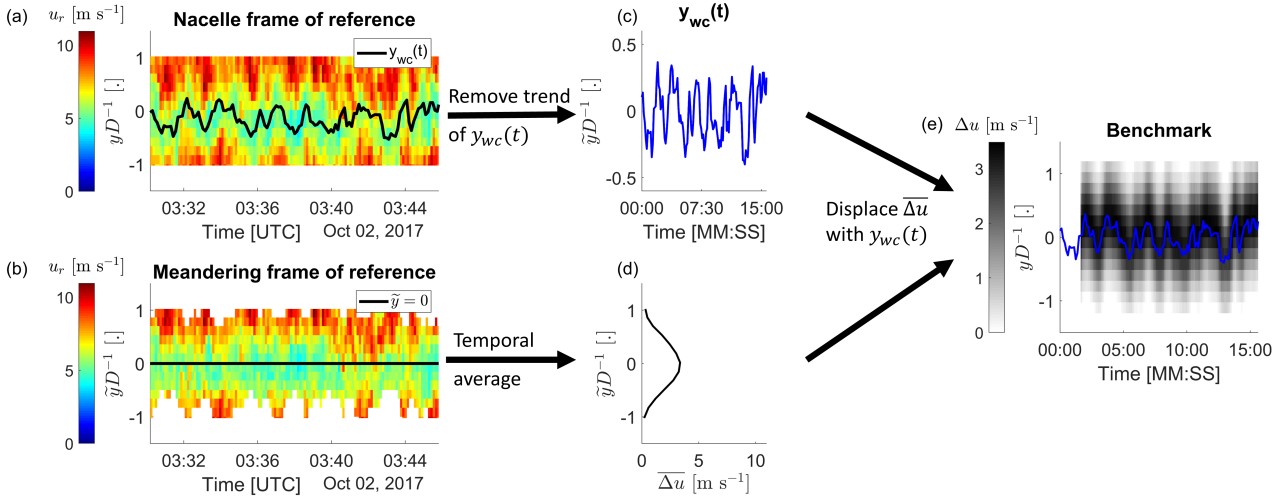

**Figure 11.** Schematic of the benchmark used to validate the DWMM. The instantaneous wake center position (a,c) and the velocity deficit in the MFOR (b,c) are combined to create a benchmark wake (e) for comparison with the DWMM.

## 3.2 Validation of the modified DWMM

The second part of the results will compare the original DWMM and the modified DWMM for the effect of wake meandering on the mean velocity deficit and the turbulence intensity. The validation is focused on the effect of wake meandering instead of the absolute values, because the absolute values can be predicted with an analytical model (e.g Qian and Ishihara, 2018) and the ability to predict the effect of wake meandering sets the DWMM apart.

### 3.2.1 Benchmark data for the model validation

Before going into the validation, a benchmark is defined. The motivation for this is that the reference data set as introduced in Sect. 2.3.2 includes the effects of a non-stationary flow (e.g. turbulence scales that are longer than the 14-minute measurement period for each case or a steadily changing wind direction for synoptic reasons) and measurement errors that are not part of the DWMM and thus should be removed for a fair comparison. The goal of the benchmark is to isolate the part of the observations that can be explained by the framework of the DWMM.

As the benchmark, we use the observed mean velocity deficit in the MFOR, which is spatially shifted based on the detrended observations of wake center positions. A schematic of the benchmark workflow is shown in Fig. 11. The mean velocity deficit in the MFOR was extrapolated with zeros outside the scanning cone of the rear-mounted Doppler LiDAR and only the part of the benchmark overlapping with the time delayed predictions of the DWMM is used. The effect of wake meandering on the mean velocity deficit and the turbulence intensity was then determined from the benchmark in the same way as for the observations (see Sect. 2.3.2 points eight and nine). The benchmark can be regarded as the best possible result that the DWMM could achieve.



A comparison of the benchmark with the observations is shown in Fig. 12. The two main differences between the observations and the benchmark for the reduction of the mean velocity deficit (Fig. 12a, b) are (i) that for a weak wake meandering
strength (small values of $\sigma(y_{wc})D^{-1}$) the observations scatter around zero, but the benchmark has strictly positive values, and (ii) that the observations reach much larger values for strong wake meandering. This causes the benchmark to have smaller slope in a linear regression and a larger intercept compared to the observations. The negative values in the observations can be explained by the method of isolating the effect of wake meandering. Random fluctuations of the detected wake center position due to measurement errors or small-scale turbulence introduce erroneous variability into the MFOR. If the wake meandering
is weak, this erroneous variability in the MFOR can be of the same magnitude as the variability in the NFOR, thus leading to values scattering around zero. The larger values in the observations for strong wake meandering can be explained with a non-stationary flow that is attributed to the effect wake meandering in the observations, but is removed from the benchmark with the detrending of $y_{wc}$. If only a mean value is removed from $y_{wc}$ instead of a linear trend, the differences in the slope of the linear regression between observation and benchmark reduces to $0.02$.

For the added turbulence intensity (Fig. 12c, d), the slope of the linear regression has better agreement between the observations and the benchmark than for the mean velocity deficit, while the offset for the intercept persists. The overall smaller differences between observations and benchmark can be explained by the lateral averaging of the quantification method, which averages strongly affected regions of the wake with more weakly affected regions further out (in contrast, the amplitude of the velocity deficit is quantified at the wake center only). Otherwise, the same arguments as for the mean velocity deficit discussed
in the previous paragraph apply.

The observations (Fig. 12a, c), which include non-stationary effects, were plotted against the trend removed wake meandering strength. If the observations are plotted against the wake meandering strength including linear trends, the correlation coefficients increase to $0.8$ and $0.9$, respectively, and the linear regression assumes similar values as reported in Brugger et al. (2022).

### 3.2.2 Prediction of the mean velocity deficit

First, we validated the predicted reduction of the mean velocity deficit by the DWMM. The DWMM using the passive tracer assumption ($\overline{u}_{hub}$ as the downstream advection velocity and $Sc_t = 1.0$) produces comparable results to the benchmark (Fig. 13a). The DWMM treating the wake as non-passive (i.e. using $\overline{u}_a$ as the downstream advection velocity and $Sc_t = 0.7$) also produces comparable results to the benchmark (Fig. 13b). Both implementations of the DWMM reproduce the trend of an decreasing
mean velocity deficit with an increasing wake meandering strength. Only when including dynamics into the validation as in Sect. 3.1, the benefit of non-passive implementation of the DWMM becomes apparent. The good results of DWMM with the passive tracer assumption are explained by the temporally averaged validation approach used here, where the errors of a too fast downstream transport and a too efficient lateral transport mostly cancel out, which might also explain why the issue was not noticed in previous studies. Overall, the good agreement to the benchmark shows that the DWMM is well parametrised
and differences to the observations can be explained by factors that are not part of the framework of the DWMM.



**Figure 12.** The reduction of the mean velocity deficit (a, b), and the added turbulence intensity (c, d) due to wake meandering as a function of the wake meandering strength at a downstream distance of $x = 5D$. The left column shows the observations and the right column shows the benchmark for the model validation. The red lines indicate a linear fit to the data.





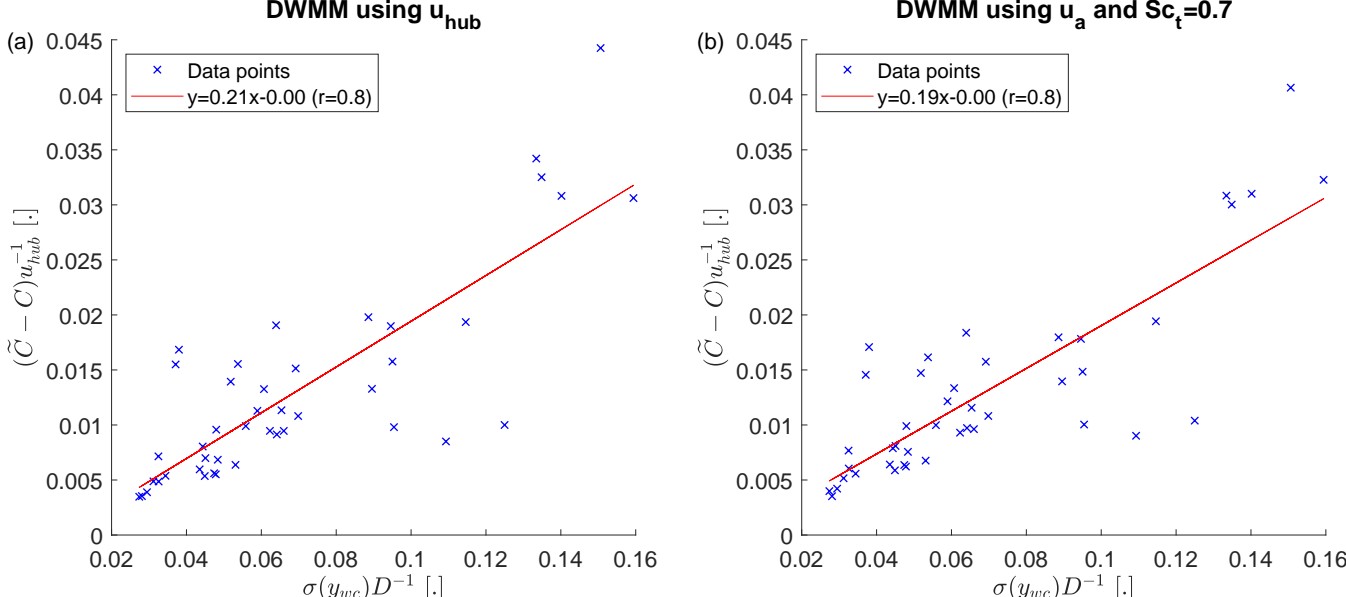

**Figure 13.** The reduction of the mean velocity deficit as a function of the wake meandering strength at a downstream distance of $x = 5D$. Panel (a) shows predictions of the DWMM using $\overline{u}_{hub}$ as the downstream advection velocity with a turbulent Schmidt number of $Sc_t = 1.0$ and panel (b) shows the DWMM using $\overline{u}_a$ with $Sc_t = 0.7$.

Recommendations in the literature for the application of the DWMM suggest a downstream transport velocity slower than the mean wind speed, but do not mention further changes to the lateral and vertical transport equations of the DWMM (Bingöl et al., 2010; Brugger et al., 2022). However, treating the wake inconsistently in the DWMM by using $\overline{u}_a$ (non-passive in the $x$ direction) and $Sc_t = 1.0$ (passive in the $y$ and $z$ direction) leads to an overestimation of the mean velocity deficit reduction compared to the benchmark (Table 1, compare slopes of the three DWMMs). This result shows that only using a slower downstream transport velocity increases the errors compared to the other two implementations of the DWMM. This is in line with our finding from the first part of the results, that only using a slower downstream transport velocity in the DWMM is not fully accounting for the non-passive nature of the wind turbine wake. Including the Schmidt number is required to accurately represent momentum transport in the wake.

### 3.2.3 Prediction of the added turbulence intensity

Lastly, the model predictions of the turbulence intensity added by wake meandering are validated. The predictions of the DWMM have good agreement with the benchmark for both the passive and non-passive implementation (Fig. 14a and Fig. 14b). The reason why both advection velocities lead to good agreement with the benchmark is the same as for the reduction of the mean velocity deficit in the previous section. Overall, the impact of the downstream transport velocity and turbulent Schmidt number on the added turbulence intensity is smaller compared to the reduction of the mean velocity deficit shown in the





**Table 1.** Correlation coefficient ($r$), slope ($a$), and intercept ($b$) of a linear regression as well as the root-mean-square error (RMSE) between the benchmark and three versions of the DWMM. The left site shows the results for the mean wake depth ($\widetilde{C} - C$) and right site shows the average turbulence intensity added by wake meandering ($\langle TI_u - \widetilde{TI}_u \rangle$). All values were computed at a downstream distance of $x = 5D$.

| | $\widetilde{C} - C$ | | | | $\langle TI_u - \widetilde{TI}_u \rangle$ | | | |
|---|---|---|---|---|---|---|---|---|
| | $r$ | $a$ | $b$ | RMSE | $r$ | $a$ | $b$ | RMSE |
| Benchmark | 1.00 | 1.00 | 0.00 | 0.00 | 1.00 | 1.00 | 0.00 | 0.00 |
| Observations | 0.79 | 1.58 | 0.00 | 0.09 | 0.82 | 1.09 | -0.95 | 1.12 |
| DWMM ($u_{hub}, Sc_t = 1.0$) | 0.86 | 1.17 | 0.02 | 0.06 | 0.85 | 0.89 | 0.29 | 0.62 |
| DWMM ($u_a, Sc_t = 1.0$) | 0.87 | 1.48 | 0.04 | 0.09 | 0.86 | 1.01 | 0.56 | 0.87 |
| DWMM ($u_a, Sc_t = 0.7$) | 0.85 | 1.06 | 0.03 | 0.05 | 0.86 | 0.85 | 0.41 | 0.58 |

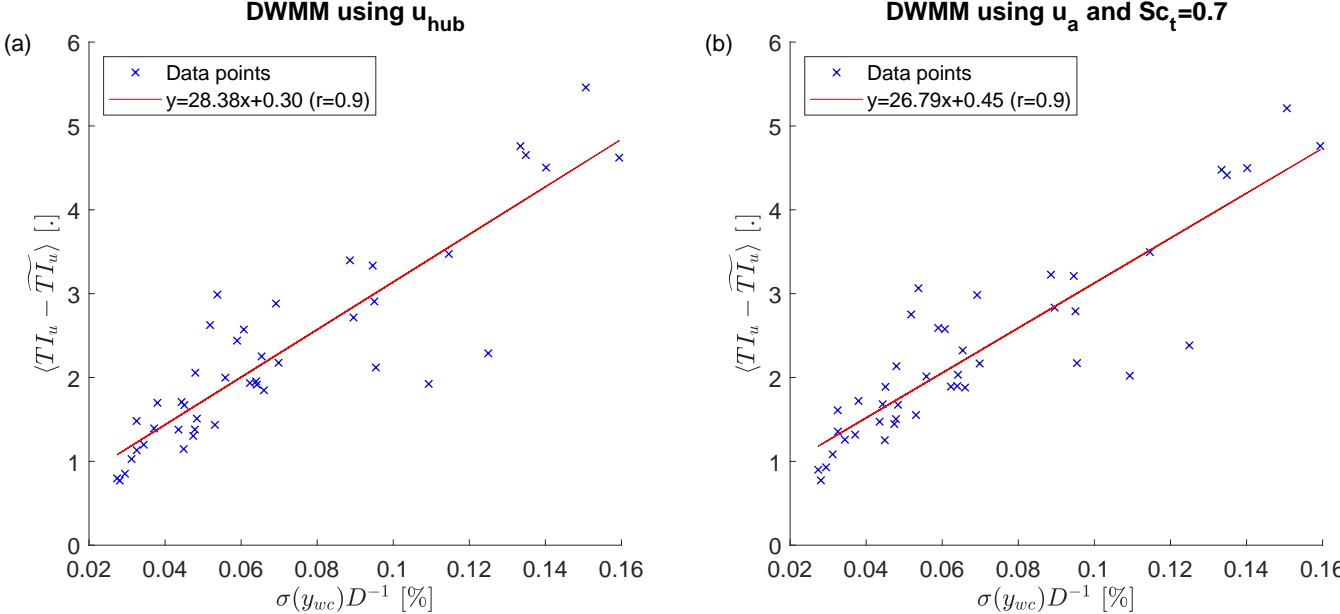

**Figure 14.** The added turbulence intensity as a function of the wake meandering strength at a downstream distance of $x = 5D$. Panel (a) shows predictions of the DWMM using $\overline{u}_{hub}$ as the downstream advection velocity with a turbulent Schmidt number of $Sc_t = 1.0$ and panel (b) shows the DWMM using $\overline{u}_a$ with $Sc_t = 0.7$.

previous section (Table 1). This is explained in part by the quantification method that includes a lateral average, which averages the most strongly affected regions of the wake with less affected regions further out.



## 4    Conclusions

A test of the existing formulation of the DWMM and a new formulation that incorporated additional physics was presented.
The test site was an isolated wind turbine in Cedar Rapids, Iowa. A Doppler lidar deployed on the nacelle of the wind turbine
scanning the velocity field of the wake at hub height was used as reference to which the models were compared. A second
Doppler lidar and the SCADA data of the wind turbine was used to initialize the wake meandering models.

The results for the instantaneous wake center position exposed an issue with the passive tracer assumption of the existing
formulation of the DWMM. The wake meandering strength had better agreement with the observation, if the mean wind
speed is used for the downstream transport, while at the same time, a better temporal agreement is reached if the downstream
transport used a special wake velocity to more accurately represent the advective transport. Analysing the transport behaviour
of the wake, we found that both the downstream transport of wake meandering as well as the lateral wake displacement showed
differences compared with the DWMM assuming a passive scalar transport. Therefore, we propose to include the turbulent
Schmidt number in the DWMM to account for the less efficient turbulent transport of momentum compared to a passive scalar
in addition to the a slower downstream transport velocity. This will also make the DWMM physically more consistent, because
the wake is considered fully non-passive with this modification, while previously it has been treated as non-passive in the
downstream direction and passive in radial direction.

A comparison of the thus modified DWMM with measurements showed that it reconciles the previously noted discrepancy
of statistics and dynamics. The DWMM model using only the more accurate downstream transport velocity had an error
increase of 64% for the mean velocity deficit reduction and 41% for the added turbulence intensity compared to the original
DWMM using the passive tracer assumption. The DWMM that included the Schmidt number in addition to the more accurate
downstream transport velocity had an error reduction for those statistics by 5% and 7%, respectively (and better temporal
agreement for the dynamics of wake meandering).

In future work, we propose a validation of our findings with a different experimental approach or through simulations to
exclude site factors or methodological biases.

*Data availability.*    The data is currently prepared to be published on Zenodo.

## Appendix A:  Equations of the Bastankhah and Porté-Agel (2016) model

The normalized mean velocity deficit of the Bastankhah and Porté-Agel (2016) model for a wind turbine aligned with the mean
wind direction is given by

$$
\frac{\Delta \overline{u}}{\overline{u}_{hub}} = A(x) \exp\left( -\frac{1}{2} \frac{y^2 + x^2}{\sigma^2} \right),
\tag{A1}
$$





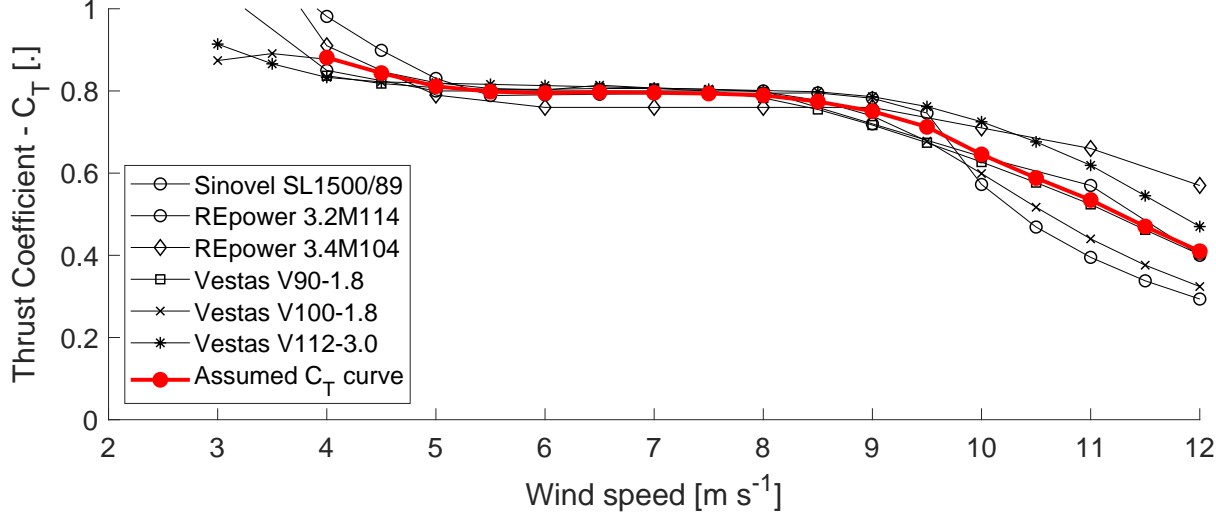

**Figure A1.** Thrust coefficient curves of six wind turbines from manufacturer data (first compiled by Abdulrahman, 2017) and the ensemble average, which is assumed as the $C_T$ curve for wind turbine at the measurement site. Figure reused from Brugger et al. (2020) without changes under the Creative Commons Attribution 4.0 License (https://creativecommons.org/licenses/by/4.0/).

where $A$ is the normalized mean velocity deficit at the wake center and $\sigma$ is the wake width. The normalized mean velocity deficit at the wake center is given by

$$A(x) = \left(1 - \sqrt{1 - \frac{C_T}{8\sigma(x)^2/D^2}}\right) \tag{A2}$$

with the thrust coefficient $C_T$. The thrust curve for this turbine model is not available to us and we will use an ensemble averaged thrust curve of several wind turbine models given there general similarity of $C_T$ between different wind turbines (Fig. A1). The $C_T$ is then chosen from the assumed thrust curve based on $\overline{u}_{hub}$. The wake width is given by

$$\sigma(x) = k^* \frac{(x - x_0)}{D} + \frac{1}{\sqrt{8}}, \tag{A3}$$

where $x_0$ is the near wake length and $k^*$ is the wake growth rate. The growth rate is computed with $k^* = 0.35 TI_u$ assuming the linear relationship between $k^*$ and the turbulence intensity found by Carbajo Fuertes et al. (2018). The near wake length $x_0$ is given by

$$x_0 = \frac{1 + \sqrt{1 - C_T}}{\sqrt{2}(\alpha^* TI_u + \beta^*(1 - \sqrt{1 - C_T}))}, \tag{A4}$$

where $\alpha^* = 2.32$ and $\beta^* = 0.154$. For the computation of the advection velocity in Eq. (14) we use $\overline{u}_{cen}(x) = \overline{u}_{hub}(1 - A(x))$ with $A(x)$ given by Eq. A2 the assumption that $A(x_0)$ is valid for $x < x_0$.





## Appendix B: Tested hypotheses for the overestimation of the wake meandering strength

The following hypotheses for the overestimation of the wake meandering strength observed in Sect. 3.1.1 were tested:

- Temporal variations of the downstream advection velocity during a 14-minute period would lead to a reduced (increased) amplitude of the wake meandering during times with faster (slower) than average advection velocity. Utilizing the outside points of PPI of the wake scanning LiDAR to gain a time series of the wind speed, we found that the effect on the predicted wake center position is too small to explain the overestimation.

- A misalignment of the wind turbine could contaminate the lateral velocity measured by the front-mounted Doppler LiDAR with contributions from the longitudinal velocity. We used the yaw angle reported in the SCADA data, and the mean wake center position within the wake scanning LiDAR's field of view to quantify the yaw misalignment of the wind turbine. The overestimation did not show any relationship to both neither in average nor trend.

- The overestimation persists if the mean instead of a linear trend is removed from $v$ and $y_{wc}$. A decrease in magnitude in
case of a removed linear trend is explained by removing the largest scales of turbulence.

- In case any remaining flow distortion of the wind turbine affecting $v(t)$ went unnoticed, range gates at a greater distance than $y = 117$ m were tested, but the overestimation persisted.

- We had the hypotheses that the onset of wake meandering is delayed due to a sheltering effect within the near wake until entrainment has reached the wake center. However, this assumption seems unrealistic based on the fact that the wake
scanning LiDAR shows wake meandering within the near wake. Testing hypotheses on the data led to a increased of the RMSE for $xD^{-1} < 5$ due to an underestimation of the wake meandering there, and small decrease of the RMSE at greater $x$.

*Author contributions.* P.B. contributed to the data curation, formal analysis, conceptualization, methodology, software, validation, visualization, and writing (original draft). F.P.-A. contributed to the conceptualization, funding acquisition, project administration, supervision, and
review and editing). C.D.M contributed to the funding acquisition, resources, data curation, investigation, and review and editing.

*Competing interests.* The authors declare that they have no conflict of interest.

*Acknowledgements.* The authors would like to thank Kirkwood Community College for their cooperation and allowing access to their wind turbine. We also extend our appreciation to Clipper Windpower for granting access to technical data on the Liberty wind turbine. This research was funded by the Swiss National Science Foundation (grant number: 200021_172538 and 200021_215288), the Swiss Federal





Office of Energy (grant number: SI/502135-01), the National Science Foundation Iowa EPSCoR (Grant No 1101284), and the Center for Global and Regional Environmental Research (CGRER), University of Iowa.



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
