# Peer review of "Improvements to the Dynamic Wake Meandering Model by incorporating the turbulent Schmidt number"

_Wind Energy Science, 2023_

## Referee Comment (RC1)

General comments:

"Improvements to the Dynamic Wake Meandering Model by incorporating the turbulent Schmidt number" makes an important contribution to the wind energy community's understanding of wake meandering. While the Dynamic Wake Meandering Model has conventionally treated the wake velocity deficit as a passive tracer advected by large-scale turbulent structures, the current manuscript addresses the shortcomings of this assumption. The connection with previous experiments indicating that momentum is transported less efficiently than scalars in turbulent wakes and the subsequent incorporation of the turbulent Schmidt number is insightful and significant for the implementation of low-cost wake prediction models. While the manuscript's contribution is novel, it lacks clarity in some parts, particularly around the discussion of the benchmark in Section 3.2. Please see below for specific comments.

Specific comments:

1. Page 1, line 18: Is it worth mentioning the theory that wake meandering is caused by bluff body vortex shedding (e.g., Medici & Alfredsson 2006)?
2. Page 2, lines 41-43: This sentence is not very clear. The paper will discuss how *what* affects the predictions of the DWMM?
3. Page 3, line 53: Even though the small-scale turbulence part of the DWMM is not used in the current study, a very brief description of how the small-scale turbulence is modeled would be nice to provide a more complete summary of the DWMM.
4. Page 4, equation 8: Please explain here how the thrust coefficient is obtained.
5. Page 5, lines 111-112: The definitions of $t$ and $\Delta T$ are not entirely clear. Is $t$ the time the wake leaves the rotor plane or the time it reaches the downstream location where wake center position is predicted?
6. Page 8, lines 173-174: What averaging period is used for the SCADA data?
7. Page 8, lines 177-178: This line further contributes to my confusion about the definition of $\Delta T$. Doesn't $\Delta T$ depend on $\bar{u}_a$, per equation 13? Does that mean the low-pass filter threshold changes for each time period?
8. Pages 8-9, lines 178-179: What is the low-pass filter threshold in terms of $D$?
9. Page 13, figure 6: It would be helpful to see the actual values for both plots in addition to the differences.
10. Page 19, section 3.2.1: What is the purpose of the extensive comparison between the observations and the benchmark? The differences discussed based on figure 12 were already described in the definition of the benchmark.
11. Page 19, lines 356-357: It's hard to compare figures 13 and 12 when they are not next to each other. Could they even be plotted on the same plot?
12. Page 19, lines 359-361: If the benchmark doesn't show the benefit of the non-passive DWMM, why is it included? Why not just compare directly with the observations?
13. Page 22, table 1: Can correlations with observations be shown in addition to (or instead of) correlations with the benchmark?

Technical corrections:

1. Page 5, line 122: Equation A2 is in appendix A, not B.
2. Page 15, line 277: Appendix B, not C.

References:

Medici, D., & Alfredsson, P. H. (2006). Measurements on a wind turbine wake: 3D effects and bluff body vortex shedding. *Wind Energy: An International Journal for Progress and Applications in Wind Power Conversion Technology*, *9*(3), 219-236.

---

## Referee Comment (RC2)

[referee-annotated manuscript omitted]

---

## Author Comment (AC1)

General comments:

"Improvements to the Dynamic Wake Meandering Model by incorporating the turbulent Schmidt number" makes an important contribution to the wind energy community's understanding of wake meandering. While the Dynamic Wake Meandering Model has conventionally treated the wake velocity deficit as a passive tracer advected by large-scale turbulent structures, the current manuscript addresses the shortcomings of this assumption. The connection with previous experiments indicating that momentum is transported less efficiently than scalars in turbulent wakes and the subsequent incorporation of the turbulent Schmidt number is insightful and significant for the implementation of low-cost wake prediction models. While the manuscript's contribution is novel, it lacks clarity in some parts, particularly around the discussion of the benchmark in Section 3.2. Please see below for specific comments.

We are grateful to Reviewer #1 for the valuable feedback and taking the time to review manuscript. Because of the reviewer's comments, we have made major changes to section 3.2, which has been completely restructured. The benchmark has been removed from the manuscript and instead we validate the DWMM directly against the observations, now. We hope that those changes have improved the clarity of Section 3.2. Please see our replies to the specific comment no. 10 below for further details.

Our replies are shown in blue and the provided line numbers refer to the revised manuscript. Additionally, a tracked-changes manuscript is provided. In response to Reviewer #2, we indicate the measurement uncertainty as error bars in many figures, now.

Specific comments:

1. Page 1, line 18: Is it worth mentioning the theory that wake meandering is caused by bluff body vortex shedding (e.g., Medici & Alfredsson 2006)?

There are two hypothesis on the origin of wake meandering: (1) large-scale turbulence of the atmospheric boundary layer flow (Larsen et al., 2008) and (2) an intrinsic shear instability of the wake leading to periodic vortex shedding (Medici and Alfredsson, 2006). Experimental support exists for both in literature. We added the above to the revised manuscript in lines 15-16.

2. Page 2, lines 41-43: This sentence is not very clear. The paper will discuss how what affects the predictions of the DWMM?

We rephrased the sentence (lines 43-45) and it now reads: "Therefore, this paper will compare the wake dynamics modelled by the DWMM to the wake dynamics observed with field measurements. Further, we investigate how differences between modelled and observed wake meandering dynamics affect the predictions of the DWMM for the effect of wake meandering on the mean velocity deficit and the turbulence intensity."

3. Page 3, line 53: Even though the small-scale turbulence part of the DWMM is not used in the current study, a very brief description of how the small-scale turbulence is modeled would be nice to provide a more complete summary of the DWMM.

The following description of the small-scale turbulence part of the DWMM was added to the manuscript (lines 56-57): "[…], and (iii) small-scale turbulence based on a homogeneous Mann (1994) turbulence field that is scaled based on the local depth of the quasi-steady velocity deficit and its radial gradient".

4. Page 4, equation 8: Please explain here how the thrust coefficient is obtained.

The thrust coefficient was selected based on the mean wind speed from the thrust curve shown in Figure A1. An explanation was added to the manuscript in lines 95-97.

5. Page 5, lines 111-112: The definitions of $t$ and $\Delta T$ are not entirely clear. Is $t$ the time the wake leaves the rotor plane or the time it reaches the downstream location where wake center position is predicted?

The variable $t$ is the time the wake reaches the downstream distance and $\Delta T$ is the time delay the wake took to get there. This was wrong in the previous version of the manuscript and has now been corrected (lines 116-117). Other implementations of the DWMM in literature have $t$ as the time when the velocity deficit leaves the rotor area, but it was more convenient for the comparison with the observations to use the timestamp of the wake measurements as the reference time.

6. Page 8, lines 173-174: What averaging period is used for the SCADA data?

Information on the averaging period has been added (lines 187-188): "Because the SCADA data has a 10-minute resolution, we use the average of a 20-minute period for the mean wind speed, which is longer than the 14-minute measurement period of the front-mounted Doppler LiDAR."

7. Page 8, lines 177-178: This line further contributes to my confusion about the definition of $\Delta T$. Doesn't $\Delta T$ depend on $\bar{u}_a$, per equation 13? Does that mean the low-pass filter threshold changes for each time period?

Yes, the filter threshold changes each period in the time domain, but it is constant in the spatial domain. We use a low-pass filter threshold that is proportional to the downstream distance, which is then transformed into a temporal threshold with $\bar{u}_a$ that can change for each time period. Using a length scale as the low-pass filter threshold rather than a time threshold is in line with other literature (e.g. Larsen et al. 2008). While the physical reasoning for the threshold in the DWMM is to isolate scales that transport the entire wake instead of deforming it, we additionally want to remove scales from the comparison that would have become de-correlated during the transport process (hence the proportionality with $x$). We state now explicitly in the manuscript that the low-pass filter threshold is proportional to the downstream distance (line 196).

8. Pages 8-9, lines 178-179: What is the low-pass filter threshold in terms of $D$?

The filter threshold is proportional to the downstream distance $x$ (added in lines 196). Specifically, the filter threshold in the spatial domain was $\beta 5D$ for Fig. 5, 7, 10, 11, and 12 as well as for Table 1. In case of Fig. 6, 8, and 9 the low-pass filter threshold varied from bin to bin according to the $x$-values on the abscissa.
We believe it is an appropriate filter threshold because it extracts the large-scale turbulence as required by the DWMM model and it also removes all scales from the comparison between the DWMM

and the observations that have become uncorrelated due to the evolution of the turbulence during the downstream transport.

9. Page 13, figure 6: It would be helpful to see the actual values for both plots in addition to the differences.

A version of Fig. 6 with the actual values is shown below. Additionally, values for the downstream distance x=5D are shown in Figure 5 of the manuscript, which explains the range of the data. The steady decrease of correlation and increase of the RMSE for x>4D is explained by a general decrease of the prediction quality with larger separations. Towards the nearest distances (x=3D), in the near wake, the velocity deficit becomes donut shaped with two peaks instead of a Gaussian profile, which can bias the centroid because they are usually asymmetrical.
However, we believe that the original Figure 6 that just shows the differences better illustrates the effect we want to highlight and additionally allows to display the variation of the differences. For that reason we did not include the below figure in the manuscript.

[Figure]

Figure 1: Correlation coefficient (a) and normalized root-mean-square error (b) between the predicted wake center position of the dynamic wake meandering model and the observed wake center position from the wake scanning Doppler LiDAR. The shaded area indicates the standard deviation. The shown values are averaged over all cases of the data set. The red line uses DWMM predictions with the mean wind speed as downstream transport velocity and the blue line shows DWMM predictions with a downstream transport velocity slower than the mean wind speed.

10. Page 19, section 3.2.1: What is the purpose of the extensive comparison between the observations and the benchmark? The differences discussed based on figure 12 were already described in the definition of the benchmark.

We have removed the benchmark from the manuscript and instead compare the DWMM directly to the observations. This led to a complete restructuring and significant shortening of Section 3.2 and required changes to the data processing that we explain in the following and summarize as a bullet-point list at the end.

In the original submission, we included the benchmark as a vehicle to explain differences between the observations and the DWMM predictions. However, it became clear to us that the benchmark failed to meet this goal and that it made the manuscript needlessly long. In the revised manuscript, we partly mitigated the need for the benchmark by removing a mean instead of a linear trend from the lateral velocity ($v$). This makes the observations and model predictions in Section 3.2 more comparable, because removing a linear trend from the observations was one of the purposes of the benchmark.

The remaining bias between predictions and observations at small wake meandering strengths is explained in the text of the manuscript without the benchmark.

Removing a mean instead of trend also affected Sect. 3.1, where the results were updated accordingly. The effects are minor and Fig. 6b and Fig. 8a are impacted the most. For Fig. 6b, the improvement of the correlation with a slower downstream advection velocity has been halved. The reason is that a steady change of the wind direction can be a major source of correlation between $v$ and $y_{wc}$, which does not depend on the advection velocity. For Fig. 8a, the predictions of the wake meandering strength using the mean wind speed have now a bias towards overestimation. However, none of those changes affected the discussion and conclusions of Sect. 3.1.

Lastly, while the benchmark was insightful to explain the differences between the model and the observations, those insights are ultimately not needed for the conclusions of the manuscript. In the new Section 3.2, we only show that the modified DWMM has similar errors as the original DWMM for the statistics (in addition to the better dynamics established in Section 3.1). Together with streamlining the discussion, this shortened the manuscript by three pages.

We list below all changes to the manuscript in response to the comment:
- Section 2.3.1, line 194: The mean instead of a linear trend is removed from $v$.
- Section 2.3.2, lines 230-231: The mean instead of a linear trend is removed from $y_{wc}$. The sentence referring to the benchmark was removed.
- Section 3.1: Fig. 5 to Fig. 10 were updated to show the new results for a removed mean and minor changes were made to the manuscript text.
    - Fig. 5a has now overall higher correlations and Fig. 5b has smaller normalized RMSE due the included linear trend. This does not affect the discussion in the manuscript text.
    - Line 260: The mentioning of the detrending was removed.
    - Fig. 6 is the most strongly affected part of Section 3.1. Including the linear trend has halved the increase in correlation shown in Fig. 6a. The reason is that a steady change of the wind direction can be a major source of correlation between $v$ and $y_{wc}$, which does not depend on the advection velocity.
    - We added a sentence stating that Fig. 6 has a larger improvement in correlation if the time series have a trend removed instead of the mean (lines 272-275).
    - Fig. 7 was updated.
    - Fig. 8 was updated. The bins for $x > 5D$ show a small positive bias for the predictions using the mean wind speed in Fig. 8a. This does not affect the finding that using $u_a$ increases the overestimation of the wake meandering strength.
    - Fig. 9 and Fig. 10 are updated as well, but changes are minor.
    - The values for the Schmidt number provided in the text have been updated (lines 335).
- Section 3.2 has been completely reworked and rewritten.
    - The section describing the benchmark has been removed from the manuscript.
    - The new Section 3.2.1 compares the predictions of the DWMM directly to the observations in Figures 11 and 12.
    - The new Section 3.2.2 investigates the impact our modification to the DWMM on the model predictions. Table 1 was streamlined in the restructuring and only shows the RMSE now.
- Conclusions

o The percentage values for the error changes were updated (lines 411 and 413). The impact of the model modification is not as pronounced with the mean removed instead of a trend, but the conclusion remains the same.

o We added another suggestion for future research (lines 414-415).

11. Page 19, lines 356-357: It's hard to compare figures 13 and 12 when they are not next to each other. Could they even be plotted on the same plot?

The model predictions and the observations are now plotted together in the new Fig. 11 and Fig. 12. The benchmark has been removed from the manuscript (see reply to comment no. 10 above).

12. Page 19, lines 359-361: If the benchmark doesn't show the benefit of the non-passive DWMM, why is it included? Why not just compare directly with the observations?

We included it initially, because the DWMM and the observations had large differences in a direct comparison, which we wanted to explain with the benchmark. However, we agree that tackling this problem in such a convoluted manner made it difficult to follow for the reader. Following our reply to specific comment no. 10, we no longer use the benchmark.

13. Page 22, table 1: Can correlations with observations be shown in addition to (or instead of) correlations with the benchmark?

The benchmark has been removed from the manuscript (see reply to comment no. 10 above) and Table 1 shows the RMSE with the observations, now. The statistics of a linear regression where removed (and can be seen in the new Fig. 11 and Fig 12 for fully-modified DWMM).

Technical corrections:

1. Page 5, line 122: Equation A2 is in appendix A, not B.

Corrected (line 128).

2. Page 15, line 277: Appendix B, not C.

Corrected (line 305).

**References**

Medici, D. and Alfredsson, P. H.: Measurements on a wind turbine wake: 3D effects and bluff body vortex shedding, Wind Energy, 9, 219–236, https://doi.org/10.1002/we.156, 2006.

Larsen, G. C., Madsen, H. A., Thomsen, K., and Larsen, T. J.: Wake meandering: a pragmatic approach, Wind Energy, 11, 377–395, 2008.

Madsen, H. A., Larsen, G. C., Larsen, T. J., Troldborg, N., and Mikkelsen, R.: Calibration and Validation of the Dynamic Wake Meandering Model for Implementation in an Aeroelastic Code, J. Sol. Energ.-T. ASME, 132, https://doi.org/10.1115/1.4002555, 041014, 2010.

---

## Author Comment (AC2)

The paper presents an interesting modification to the DWM model, that moves away from the assumption that the wake is transported as passive scalar and instead is akin to momentum transport, which is less efficient. The authors are using lidar wake measurements to motivate their modification. Overall the paper is clear, well structured and written, but fails to make full use of the measurements available.

Whilst the paper is sufficiently detailed with respect to the modelling choices and underlying procedures the validation approach remains somewhat unclear and is completely lacking any uncertainty quantification (measuring the lateral component with a lidar for instance should have large uncertainty). Spatially and temporal varying measurements of the wake need more rigorous treatment than stationary data if they are meant to be useful in the context of validating a dynamic wake model. There is temporal and spatial variation in the reference data and the measurement uncertainties need to be propagated to the derived quantities like the wake centre location. They should also propagate the input uncertainties through the DWM model and then compare with the observations. The authors need to perform validation under uncertainty to clearly demonstrate that their is statistically significant improvement from their modification of the DWM model. The linear regression lines shown throughout the submission are not sufficient. There are plenty of previously published studies using complex lidar measurements for model validation the authors could refer to for inspiration. The scientific impact of the submission will be much greater once all uncertainties are accounted for.

We are grateful to Referee #2 for the provided feedback and for reviewing the manuscript. We acknowledge that we have not treated the measurement uncertainty with the needed scientific rigor, which we addressed in the revised manuscript. Briefly summarized, we estimated the uncertainty of our measurement data and propagated it to the model predictions and the derived wake quantities. We detail this and the changes to the manuscript in the following. Also, please note that we implemented extensive changes to section 3.2 of the manuscript due to the feedback of Referee #1.

We make the following assumptions for the initial measurement uncertainty:
- The wake-scanning lidar used a signal-to-noise ratio (SNR) threshold of -14 dB, 3000 averaged pulses per estimate, and six points per range gate. For those settings, we estimate the uncertainty of the radial velocity as 0.3 m/s (Pearson et al. 2009, Eq. (2) therein). This uncertainty applies to the SNR threshold, but our data points have a better SNR than -14 dB and therefore this is an upper bound for the uncertainty. We previously used a SNR threshold of -17 dB, but we increased it for the revised manuscript. A technical report by Newsom and Krishnamurthy (2022) shows uncertainties of 0.1-0.2 m/s for six different Halo Photonics Stream lidars at an SNR of -14 dB experimentally, which is in line with our estimate for the uncertainty.
- For the spatial uncertainty of the wake-scanning lidar, we assume that it is equal to the azimuth distance travelled by the scanner head during a measurement. The assumed uncertainty of the azimuth is therefore 2° for our PPI scans. For spatial errors due to tower bending, see the last part of our reply.
- The measurements of the lateral velocity from the forward-mounted lidar have a much higher SNR than the -14dB threshold due to short measurement distance. Using Eq. (2) of Pearson et al. 2009 and the recorded SNR values leads to a theoretical uncertainty that is always smaller than 0.02 m/s across the data set, which is lower than the velocity resolution of instrument (0.038 m/s). Therefore, we use the velocity resolution as the uncertainty for the lateral velocity.

- We found a root-mean-square error (RMSE) of 0.45 m/s between the mean wind speed from the SCADA data and the mean wind speed from upstream stares parallel to the rotor axis of the front-mounted Doppler lidar. The upstream stares were mentioned in the manuscript, but their data had not been used so far. The RMSE includes errors resulting from the spatial separation of those measurements, different time periods averaged, and a bias that likely originates from the induction zone. If we remove the bias before computing the RMSE it reduces to 0.25 m/s, which is used as uncertainty of the mean wind speed for the error propagation.

Based on the above uncertainties, we employed a Monte Carlo method to estimate the propagated uncertainty. We created 100 resamples of the measurement data of a given case by adding random fluctuations drawn from a normal distribution with a standard deviation equal to the above estimated uncertainties. In case of the azimuth uncertainty, we used a uniform distribution across an interval equal to the uncertainty. This procedure was applied to the measurement values of the lidar radial velocities, the lidar azimuth readings, and the SCADA wind speed. We then recomputed the wake quantities and the DWM model predictions for each resample. Lastly, the propagated uncertainty was quantified as the RMSE between the original result and the results of the 100 resamples. This procedure was applied to each of the 43 cases of our data set. If results are normalized in a figure, we apply the error propagation rules for a division as a last step.

In case of the DWMM, the Monte Carlo approach was implemented in two stages. First, we estimated the uncertainty of $y_{pre}$ with the Monte Carlo approach based on the uncertainties of $v$ and $\bar{u}_{hub}$. Then, we estimated the uncertainty of the mean velocity deficit and the added turbulence intensity with a second Monte Carlo approach based on the uncertainties of $y_{pre}$ and $\bar{u}_{hub}$.

To present the uncertainty in the manuscript, we made the following additions:
- The measurement uncertainties of the instruments are stated in the methods sections, when we introduce the measurement setup (lines 161-165, 176-179, and 189-190).
- Subsection 3.2.2 was added that introduces the method of error propagation (lines 237-247).
- The propagated measurement uncertainty is displayed as error bars of the data points in Fig. 5, 10, 11, and 12. We split Fig. 10 into two panels for clearer visibility with the error bars and swapped x-axis and y-axis to be more intuitive.
- Additionally, we provide for those figures the confidence interval of the linear regression to show its statistical uncertainty due the scatter of the data.
- We did not include the measurement uncertainty in Fig. 6 and 8, because the existing error bars in those figures show the variation due changes of the environment conditions that we deem more important here (Fig. 6 and 8 show the same quantities as Fig. 5 and 10 and the propagated measurement uncertainty can be seen there).

Let us now focus on the reviewer's comment on the uncertainty of lateral velocity measurements with a Doppler lidar. Even when we assume a higher uncertainty for the lateral velocity, the low-pass filter applied to the data leads to substantial temporal averaging, which reduces the propagated uncertainty for the DWM model predictions (we tested an uncertainty of 0.2 m/s and the errors of the DWM model were acceptable). For the predictions of the DWM model, the uncertainty of the mean wind speed proved to be a more substantial error source.

Overall, the results of the uncertainty propagation are consistent with the behavior of the data points. The error bars in Fig. 10-12 are not large enough to explain a strong scattering of the

data points, which is consistent with the high correlation coefficients reported ($r \geq 0.8$). Figure 5 has a lower correlation, which is consistent with comparatively larger error bars there.
Further, the error analysis shows that the errors of the DWM and the observed wake quantities are not large enough to invalidate the results in our opinion. The modification of the DWM model led to a significant improvement, which can be seen from the confidence bounds of the linear fit in Fig. 10a that do not cover the identity, while they do in Fig. 10b.

We want to close by discussing tower bending. Even though the instrument was leveled before the campaign, a tilting of the instrument due to tower bending (a "nodding" like fore-aft movement of the tower) cannot be excluded. We do not have adequate support measurements to quantify this properly (the lidar's internal pitch and roll sensors seem very noisy even if the instrument is on solid ground).
However, based on maximum tower top displacements for above rated wind speeds found in the literature, we can estimate an upper bound for a tilting of the lidar beams. A maximum tower top displacement ($\Delta x$) of 0.2 m was given in Bossanyi (2003). Two further estimates found in the grey literature provided similar values ($\Delta x = 0.2$ m in a technical report by Hooft et al. (2003) for a turbine with a hub height of 92 m and $\Delta x = 0.12$ m by Mate Jelavic et al. (2007) in conference proceedings). To estimate the effect on the lidar beam, we assume that the tower is stiff and compute the beam misalignment with $tan^{-1}(\Delta x/z_{hub})$, which results in a maximum beam misalignment of $0.15°$ for $\Delta x = 0.2$ m. The corresponding vertical displacement of the lidar beam is $\Delta z = 0.72$ m at $x = 3D$ and $\Delta z = 1.94$ m at $x = 8D$.
We expect the effect on the wake center position to be small assuming a regular shape of the wake (i.e. no branching). The effect on the mean velocity deficit should be small as well, because the wind shear over this height range should be small compared to the velocity deficit of the wake.

**References**

Pearson, G., Davies, F., and Collier, C.: An analysis of the performance of the UFAM pulsed Doppler lidar for observing the boundary layer, J. Atmos. Ocean Tech., 26, 240–250, https://doi.org/10.1175/2008JTECHA1128.1, 2009.

Newsom, RK, and Krishnamurthy, Raglavendra. Doppler Lidar (DL) Instrument Handbook. United States: N. p., 2022. Web. doi:10.2172/1034640.

E.A. Bossanyi (2003): Wind Turbine Control for Load Reduction. Wind Energy, 6:229-244. DOI: 10.1002/we.95.